# Osteopontin Predicts Three-Month Outcome in Stroke Patients Treated by Reperfusion Therapies

**DOI:** 10.3390/jcm9124028

**Published:** 2020-12-13

**Authors:** Elena Meseguer, Devy Diallo, Julien Labreuche, Hugo Charles, Sandrine Delbosc, Gabrielle Mangin, Linsay Monteiro Tavares, Giuseppina Caligiuri, Antonino Nicoletti, Pierre Amarenco

**Affiliations:** 1Department of Neurology and Stroke Center, APHP Bichat Hospital, F-75018 Paris, France; hcharles.bichat@gmail.com (H.C.); linsay.monteiro.biostat@gmail.com (L.M.T.); pierre.amarenco@aphp.fr (P.A.); 2LVTS, Inserm U1148, Université de Paris, F-75018 Paris, France; devy.diallo@gmail.com (D.D.); sandrine.delbosc@inserm.fr (S.D.); gabrielle.mangin@inserm.fr (G.M.); giuseppina.caligiuri@inserm.fr (G.C.); antonino.nicoletti@inserm.fr (A.N.); 3Department of Biostatistics Université de Lille, CHU Lille, ULR 2694-METRICS: Évaluation des Technologies de Santé et des Pratiques Médicales, F-59000 Lille, France; julien.labreuche.chru@gmail.com

**Keywords:** acute stroke, biomarkers, assessment, outcomes, thrombolytic therapy, thrombectomy

## Abstract

Establishing a prognosis at hospital admission after stroke is a major challenge. Inflammatory processes, hemostasis, vascular injury, and tissue remodeling are all involved in the early response to stroke. This study analyzes whether 22 selected biomarkers, sampled at admission, predict clinical outcomes in 153 stroke patients treated by thrombolysis and mechanical endovascular treatment (MET). Biomarkers were related to hemostasis (u-plasminogen activator/urokinase (uPA/urokinase), serpin E1/PAI-1, serpin C1/antithrombin-III, kallikrein 6/neurosin, alpha 2-macroglobulin), inflammation[myloperoxidase (MPO), chemokine ligand 2/monocyte chemoattractant protein-1 chemokine (CCL2/MCP-1), adiponectin, resistin, cell-free DNA (cDNA), CD40 Ligand (CD40L)], endothelium activation (Vascular cell adhesion protein 1 (VCAM-1) intercellular adhesion molecule 1 (ICAM-1), platelet endothelial cell adhesion molecule 1 (CD31/PECAM-1)], and tissue remodeling (total cathepsin S, osteopontin, cystatin C, neuropilin-1, matrix metallopeptidase 2 (MMP-2), matrix metallopeptidase 3 (MMP-3), matrix metallopeptidase 9 (MMP-9), matrix metallopeptidase 13 (MMP-13)]. Correlations between their levels and excellent neurological improvement (ENI) at 24 h and good outcomes (mRS 0–2) at three months were tested. Osteopontin and favorable outcomes reached the significance level (*p* = 0.008); the adjusted OR per SD increase in log-transformed osteopontin was 0.34 (95%CI, 0.18–0.62). The relationship between total cathepsin S and MPO with ENI, was borderline of significance (*p* = 0.064); the adjusted OR per SD increase in log-transformed of total cathepsin S and MPO was 0.54 (95%CI, 0.35–0.81) and 0.51 (95%CI, 0.32–0.80), respectively. In conclusion, osteopontin levels predicted three-month favorable outcomes, supporting the use of this biomarker as a complement of clinical and radiological parameters for predicting stroke prognosis.

## 1. Introduction

The Global Burden of Disease estimated that one person in four aged 25 years will have a stroke in the rest of her/his life [1]. Among them, ischemic stroke represents 80%. Stroke accounts for almost 5% of all disability-adjusted life-years and 10% of all deaths worldwide [1]. These numbers make stroke a major healthcare problem all over the world.

Acute ischemic stroke (AIS) treatments include intravenous (iv) thrombolysis, and mechanical endovascular treatment (MET) in the case of accessible arterial occlusion [2]. These procedures allow arterial recanalization and reperfusion of the ischemic area and improve the overall three-month clinical outcome [3], with a risk of hemorrhagic transformation. Time-to-treatment and time-to-reperfusion are key determinants of the efficacy of thrombolysis and MET [4]. In addition, NIHSS at admission [5], age [6], diabetes mellitus [7,8], and/or hyperglycemia [7]_,_ the presence of arterial collaterals [9], size of the infarction [10], and blood-brain barrier disruption after treatment [11] also determine the treatment efficacy. Even if these clinical and radiological parameters are well known, predicting the outcome in individual patients solely based on these parameters is challenging for clinicians. It has been proposed that measuring blood biomarkers covering distinct pathways, including inflammation, endothelial function, matrix remodeling, renal, cardiac, and immune functions, could improve prediction performance [12]. Herein, we aimed at evaluating a panel of 22 biomarkers focused on cellular and molecular early processes closely associated with stroke, namely, hemostasis, inflammation, endothelium activation, and tissue remodeling. We analyzed the value of these biomarkers at admission for predicting the outcome of patients treated by thrombolysis and/or Endovascular Treatment (EVT). We tested the correlation between the levels of these blood biomarkers with clinical evolution at 24 h and three months post-stroke.

## 2. Experimental Section

### 2.1. Patients

We conducted a prospective study with patients admitted for ischemic stroke who received thrombolysis and/or MET within 6 h of symptom onset at Bichat Hospital, Paris, France. Only patients arriving within working hours of the research nurse were included in the study. Plasma samples were taken from all patients at arrival, before any treatment was given. We collected data from each patient on demographics, risk factors, previous medications for stroke, baseline National Institutes of Health Stroke Scale (NIHSS) [13], and blood pressure at admission. All patients had brain imaging to support the diagnosis. Patients with ischemic stroke were treated with iv recombinant tissue plasminogen activator (rtPA), and in the case of documented arterial occlusion EVT, including intra-arterial (IA) thrombolysis +/− MET was performed.

### 2.2. Data Collection and Definitions

Data were collected using a structured questionnaire. Smoking was classified as current (any smoking within the past six months), former (more than six months since cessation), or never smokers. Hypertension and hypercholesterolemia were defined by treatment history. Subjects were classified as diabetic when treated for type 1 or type 2 diabetes. The stroke severity was assessed using the NIHSS score at admission. The arterial status of the occluded artery was monitored with conventional angiography during the endovascular procedure and measured using the Thrombolysis in Cerebral Infarction (TICI) score [14]. All patients underwent a computed tomography (CT) or magnetic resonance imaging (MRI) scan 24 h after treatment onset, to assess hemorrhagic complications. Intracranial hemorrhage was defined and classified according to the European Co-operative Acute Stroke Study III (ECASS-III) trial [15]. Functional outcomes were assessed at three months using the modified Rankin Scale (mRS) [16] during face-to-face interviews or via telephone calls by a senior vascular neurologist (E.M) certified for mRS evaluation.

All participants provided informed consent. This study was performed according to the principles of the Declaration of Helsinki (1964). Informed consent was obtained from the patient or surrogate, and the research protocol was approved by the Ethics Committee from the Ambroise Pare Hospital, on 23rd September 2008, code ID RCB:2008-A00966-49 and 05th January 2010, code ID RCB: 2009-A01165-52, Paris, France.

### 2.3. Outcome Measurements

A favorable outcome was defined as an mRS score of ≤2 at 90 days. An excellent favorable clinical outcome was defined as an mRS score of ≤1 at 90 days. Early neurological improvement (ENI) was defined as an improvement of 4 or more points on this scale (compared with baseline) 24 h after thrombolytic/EVT treatment or NIHSS of 0 24 h after thrombolytic/EVT treatment.

In the case of EVT, the recanalization rate was measured using the TICI at the end of the endovascular procedure. Successful recanalization was defined as TICI 2B or 3 [14]. Hemorrhagic transformation was defined as the appearance of hemorrhagic transformation at 24 h on CT or MRI. Symptomatic hemorrhagic transformation was defined according to the ECASS-III definition as the presence of any hemorrhage on the follow-up CT/MRI imaging scan associated with an increase of 4 points in NIHSS score or death within 7 days [15].

### 2.4. Blood Biomarkers

Patients were sampled within 6 h after stroke onset and before any reperfusion treatment was given. We selected 22 biomarkers according to their main functions in ischemic stroke:

**Hemostasis**: u-plasminogen activator/urokinase (uPA)/urokinase, serpin E1/PAI-1, serpin C1/antithrombin-III, kallikrein 6/neurosin, alpha 2-macroglobulin. **Inflammation**: Myloperoxidase (MPO), chemokine ligand 2/monocyte chemoattractant protein-1 chemokine (CCL2/MCP-1), adiponectin, resistin, cell-free DNA (c-DNA), CD40 Ligand (CD40L). **Endothelium activation**: Vascular cell adhesion protein 1 (VCAM-1), intercellular adhesion molecule 1 (ICAM-1), platelet endothelial cell adhesion molecule 1 (PECAM-1 or CD31). **Tissue remodeling**: Total cathepsin S, osteopontin, cystatin C, neuropilin-1, MMP-2, matrix metallopeptidase 2 (MMP-2), matrix metallopeptidase 3 (MMP-3), matrix metallopeptidase 9 (MMP-9), matrix metallopeptidase 13 (MMP-13).

The concentration of biomarkers, except c-DNA, was measured in plasma, collected on EDTA tubes, by using commercials human Magnetic Luminex assay kits (reference LXSAHM, R&D system Bio-Techne) according to the manufacturer’s instruction. Briefly, 50 µ l of diluted plasma or standard was incubated with magnetic beads (pre-coated with analyte-specific antibodies). After different steps of washing, samples or standards were incubated with biotinylated antibodies specific to the analytes of interest. Following a wash to remove any unbound biotinylated antibody, streptavidin-phycoerythrin conjugate (Streptavidin-PE), which binds to the biotinylated antibody, was added to samples or standards, and after final washes, the fluorescence intensity of the magnetic beads was detected on a Bio-Plex 200 system (Bio-rad). The minimum detection limit in ng/mL is for uPA/urokinase 0.12, serpin E1/PAI-1 0.02, serpin C1/antithrombin-III 2.84, total cathepsin S 0.09, CCL2/MCP-1 0.03, CD31/PECAM-1 0.68, CD40L 0.21, ICAM-1 6.89, kallikrein 6/neurosin 0.28, MMP-3 0.08, MMP-13 0.12, osteopontin 1.8, resistin 0.04, VCAM-1 7.80, alpha 2-macroglobulin 1.66, adiponectin 0.95, cystatin C 0.66, MMP-9 0.13, neuropilin-1 0.08, MMP-2 0.32, MPO 0.13.

The c-DNA concentration was determined in plasma EDTA, using QuantitTM PicogreenH dsDNA Reagent (Invitrogen). Briefly, 10 µL of samples and Lambda DNA standard (1 ng/mL–1 mg/mL) were diluted in TE buffer (200 mM Tris-HCl, 20 mM EDTA, pH 7.5, 100 mL final) before addition of 100 µL PicogreenH dsDNA reagent. After mixing, and incubation for 5 min at room temperature in the dark, the fluorescence was measured using a microplate reader (excitation 480 nm, emission 520 nm).

### 2.5. Statistics

Quantitative variables are expressed as mean (standard deviation) in the case of normal distribution or median (interquartile range) otherwise. Categorical variables are expressed as numbers (percentages). The normality of distributions was assessed using histograms and the Shapiro-Wilk test. To assess the selection bias, baseline characteristics were described according to the included and non-included patients, and the magnitude of the between-group differences was assessed by calculating the absolute standardized differences; an absolute standardized difference <0.2 was interpreted as a small difference [17]. Baseline characteristics were further described and compared according to the pre-specified primary outcome (favorable outcome) using the Chi-square test or Fisher’s exact test for categorical variables, and Student’s *t*-test or Mann-Whitney U-test for quantitative variables according to the normality of distributions. We firstly assessed the associations of blood biomarkers with favorable, excellent, and ENI outcomes in bivariate analyses by using Student’s *t*-test on log-transformed (to reduce the skewness of distribution) for all blood biomarkers (except for Alpha2 macroglobulin). Associations of blood biomarkers with each outcome were further investigated in multivariate logistic regression analysis adjusted for the following pre-specified confounders: age, admission NIHSS, documented occlusion (none vs. Internal Carotid Artery (ICA) occlusion vs. others arterial occlusion), and AIS treatments. Strength of associations was assessed by calculating the adjusted odds ratios (ORs) and theirs 95% confidence intervals (CIs) per one SD increase in blood biomarkers. Log-linearity assumptions were checked using restricted cubic spline functions [18]. *p*-values for bivariate and multivariate analyses were corrected for multiple comparisons (22 biomarkers tested) by using the Holm-Bonferroni method. Statistical testing was done at the two-tailed α level of 0.05. Data were analyzed using the SAS software package, release 9.4 (SAS Institute, 100 SAS Campus Drive Cary, NC 27513-2414, Cary, NC, USA).

## 3. Results

### 3.1. Study Sample

From January 2009 to June 2013, 618 patients with AIS within 6 h of symptom onset were consecutively treated by AIS treatment (thrombolysis and/or EVT) at our center. Of these patients, 151 patients had a blood sample taken just prior to acute treatment with 90-day mRs evaluation and were included in the present study (Figure 1).

The median age of the study sample was 73 years (IQR, 58 to 83), and 51% of them were men. The median NIHSS was 13 (IQR, 5 to 18), and 31% of patients had no documented occlusion. 68 patients were treated by IV thrombolysis, 55 by an IV + EVT approach (ia thrombolysis *n* = 21 and ia thrombolysis + adjunctive MET, *n* = 34), and 28 by an exclusively EVT (ia thrombolysis, *n* = 8; MET, *n* = 15; ia thrombolysis + adjunctive MET, *n* = 5). Baseline characteristics for included and excluded patients are available in Appendix A; except for diabetes (less frequent in included patients), admission NIHSS (higher in included patients), documented occlusion (more frequent in included patients), and AIS treatment (more frequent use of an endovascular approach in included patients, especially MET), baseline characteristics were well balanced (standardized difference <20%).

In study sample, 72 patients (47.7%; 95%CI, 39.7–55.7) had an ENI, 64 (42.4%; 95%CI, 34.5–50.3) had an excellent functional outcome and 80 (53.0%; 95%CI, 45.0–60.9) had a favorable outcome. Baseline characteristics of patients with and without favorable outcomes (pre-specified as primary outcomes) are reported in Table 1. Patients with favorable outcomes were significantly younger, had less often pre-stroke mRS >1, a lower glucose level at admission, a lower NIHSS score at admission, less often documented occlusion, and were consequently more often treated by IV fibrinolysis alone.

### 3.2. Blood Biomarkers

Distribution of the 22 biomarkers tested (five for hemostasis, six for inflammation, three for endothelial activation, and eight for tissue remodeling) are reported in Appendix A.

#### 3.2.1. Hemostasis Biomarkers and Outcomes

As shown in Table 2, Table 3 and Table 4, in bivariable analysis corrected for multiple comparisons, none of the hemostasis biomarkers were associated with any outcomes. Similar results were found in a multivariable analysis adjusted for pre-specified confounders (age, admission NIHSS, documented occlusion, and AIS treatment).

#### 3.2.2. Inflammation Biomarkers and Outcomes

Of the six inflammation biomarkers, only MPO was associated with each study outcome in the bivariable analysis (p corrected for multiple comparisons <0.01). MPO was lower in patients with favorable outcomes than those without favorable outcomes (median (IQR), 38 (28–50) vs. 51 (37–74)). Similarly, an increased MPO level was associated with a reduced likelihood of excellent (Table 3) and ENI outcomes (Table 4). In a multivariable analysis adjusted for pre-specified confounders, MPO remained associated with reduced likelihood of each outcome before, but not after correction for multiple comparisons; the adjusted OR per one SD increase in log-transformed MPO was 0.51 (95%CI, 0.32 to 0.89) for a favorable outcome, 0.60 (95%CI, 0.37 to 0.97) for an excellent outcome, and 0.51 (95%CI, 0.32 to 0.80) for ENI.

#### 3.2.3. Endothelial Activation Biomarkers and Outcomes

In bivariable analysis corrected for multiple comparisons, only VCAM-1 was significantly associated with a favorable and excellent outcomes, but not with ENI. Similar to MPO, increased VCAM-1 level was associated with a reduced likelihood of favorable and excellent in multivariate analysis before, but not after correction for multiple comparisons (Table 2 and Table 3); the adjusted OR per one SD increase in log-transformed VCAM-1 was 0.53 (95%CI, 0.31 to 0.90) for a favorable outcome and 0.54 (95%CI, 0.33 to 0.89) for an excellent outcome.

#### 3.2.4. Tissue Remodeling Biomarkers and Outcomes

Of the eight tissue remodeling biomarkers, a favorable outcome was significantly associated with lower osteopontin, and neuropilin-1 levels in bivariable analysis corrected for multiple comparisons (Table 2); a similar association (but not significant) was observed for total cathepsin S and cystatin C. Regarding other outcomes, total cathepsin S, osteopontin and neuropilin-1 were associated with reduced likelihood of excellent outcomes (Table 3), and only total cathepsin S was associated with reduced likelihood of ENI (Table 4). In a multivariable analysis, total cathepsin S and osteopontin were associated with reduced likelihood of functional outcomes, but only the association between osteopontin and the favorable outcome reached the significance level after multiple comparison correction (*p* = 0.008); the adjusted OR per SD increase in log-transformed osteopontin was 0.34 (95%CI, 0.18 to 0.62) for a favorable outcome and 0.49 (95%CI, 0.29 to 0.81) for an excellent outcome. The association of total cathepsin S with ENI was of borderline significance in a multivariable analysis corrected for multiple comparisons with an OR per SD increase in log-transformed total cathepsin S of 0.54 (95%CI, 0.35 to 0.81).

#### 3.2.5. Osteopontin and Favorable Outcomes According to AIS Treatment

As shown in Figure 2, there is no evidence of heterogeneity in association with osteopontin levels and a favorable outcome, according to AIS treatment. In patients under exclusively IV thrombolysis and those treated by a combination of IV thrombolysis and EVT; osteopontin was related to poor outcomes (*p* = 0.014 and *p* = 0.007, respectively). In those under EVT (including IA thrombolysis and/or MET), there was no difference (*p* = 0.13), but only six patients with favorable outcomes were included in the analyses (Figure 2A). Osteopontin levels were also related to favorable outcomes in patients who received MET (*p* = 0.026) and those who not (*p* = 0.001) (Figure 2B)**.**

## 4. Discussion

Previous studies in ischemic stroke have correlated blood biomarkers to clinical presentation [19], outcomes [12], and hemorrhagic transformation [20]. In our study, low levels in plasma EDTA of circulating osteopontin, a tissue remodeling biomarker, predicted favorable outcomes at three months in adjusted and unadjusted analyses. Moreover, cathepsin S and MPO levels in plasma EDTA were predictors in non-adjusted analyses of ENI.

Patients with lower levels of osteopontin did better at three months than those with higher levels. It was the only biomarker that predicted favorable outcomes at three months in adjusted and unadjusted analyses, and it also predicted excellent outcomes in unadjusted analyses. These results were independent of the revascularization treatment. Osteopontin is an acidic phosphoglycoprotein that was first described as being secreted by transformed cells isolated from bone [21,22]. It is upregulated in response to injury, stress, and inflammation in diverse cells [23]. This protein is a potent mediator in the development and atherosclerosis progression [24], and is involved in hemostasis, angiogenesis, wound healing, and immune responses [25,26,27,28]. In the brain, OPN is expressed constitutively and is upregulated upon experimental cerebral ischemia [29,30], where it has been shown to play a role in remodeling processes by modulating macrophage recruitment, matrix repair, astroglial cell migration, and gliosis. In accordance with our results, a previous study in patients treated by IV thrombolysis has found that high initial levels of osteopontin were related to bad outcomes at three months [31]. Moreover, another study has shown that high serum levels of osteopontin seven days after ischemic stroke, were positively correlated with cerebral infarction volume and bad neurological status at 3-month [32]. High osteopontin levels were also related to stroke recurrence in the SPARCL Trial [33] and bad outcomes, in the case of stroke recurrence in this trial (unpublished data). All these data points at osteopontin as a key informative biomarker in the case of ischemic stroke.

We also found that circulating levels of cathepsin S were associated with reduced likelihood of excellent outcomes and of ENI, but it must be noted that statistical significance was lost after multiple comparison correction. Cathepsins are proteases that promote the degradation of damaged proteins in the lysosome, and they are also secreted extracellularly in a variety of pathological processes, including arthritis, cancer, and cardiovascular disease [34]. Specifically, cathepsin S has been implicated in elastin degradation in cardiovascular disease, and high levels of cathepsin S have been described in the atheroma plaque [35]. Experimental animal studies have shown that cathepsin S promotes endothelial cell invasion and neovessel growth [36]. To the best of our knowledge, the role of cathepsin S in ischemic stroke has not yet been studied. Our study suggests that this biomarker could be informative for the clinical evolution at 24 h, but more studies are needed to confirm this hypothesis.

MPO levels were also borderline as a predictor of a reduced likelihood of ENI. MPO is produced mostly by polymorphonuclear neutrophils [37]. MPO is stored in cytoplasmic membrane-bound azurophilic granules, which can be secreted out to the extracellular space by degranulation or exocytosis upon oxidative stress and different inflammatory responses. Myeloperoxidase is the only type of peroxidase that uses H2O2 to form different hypohalous acids [38]. Through its activities, MPO increases tissue injury [39]. In an experimental model of cerebral ischemia-reperfusion in rats, MPO levels were found to be significantly increased in the cerebral cortex 24 h after reperfusion, and brain MPO activity correlated with the appearance of neutrophils. Neutrophil depletion prevented increased MPO activity and reduced infarct size [40]. MPO knockout mice subjected to the same stroke model also showed a similar reduction in the final lesion volume [41]. Human studies showed that patients with ischemic stroke display higher serum levels of MPO compared to healthy subjects [42], and higher levels of MPO are related to higher NIHSS [42] and mortality [43]. These data might indicate an early deleterious involvement of neutrophils, since high levels of circulating MPO level is a marker of increased neutrophil activity. In line with this hypothesis, higher neutrophil counts before thrombolysis for cerebral ischemia predicted worse outcomes [44]. Altogether, these reports and our study indicate that circulating MPO level taken as a surrogate marker of neutrophil activity is a useful biomarker for stroke prognosis.

Few limitations derive from the design of the present study. Patients were treated by IA thrombolysis and first-generation thrombectomy devices. Nowadays, devices display better performance, and higher levels of recanalization are achieved. We did not include patients treated by antithrombotic treatment; the interest of these biomarkers in this population is unknown. Samplings were from a single-center, and they were collected in a limited number of patients. These results should be taken as preliminary and exploratory, and need to be reproduced in larger, multicenter cohorts to be generalizable and to ultimately develop a point of care device. Yet, the objective of this study was to analyze the predictive value of biomarkers that had been only scarcely studied in the setting of patients treated with thrombolysis.

## 5. Conclusions

In conclusion, osteopontin predicted favorable outcomes at three-month, while cathepsin S and MPO were related to ENI in unadjusted analysis. No biomarker related to hemostasis or endothelial activation biomarker predicted ENI or three-month outcome. We did not find any biomarker related to excellent outcomes. Larger studies are warranted to confirm the role of these biomarkers as predictors of stroke outcome.

## Figures and Tables

**Figure 1 jcm-09-04028-f001:**
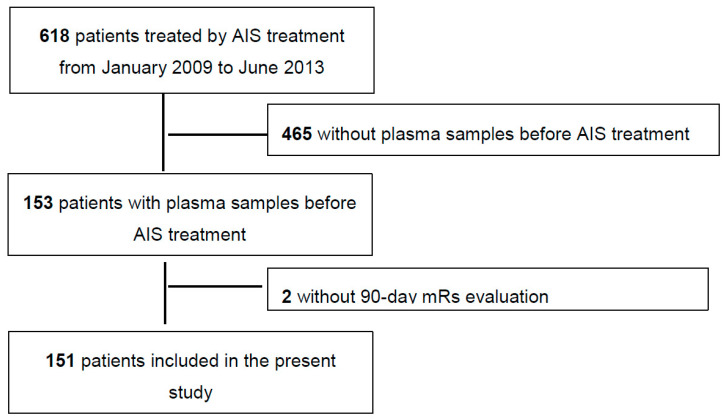
Study flow-chart. AIS = Acute ischemic stroke.

**Figure 2 jcm-09-04028-f002:**
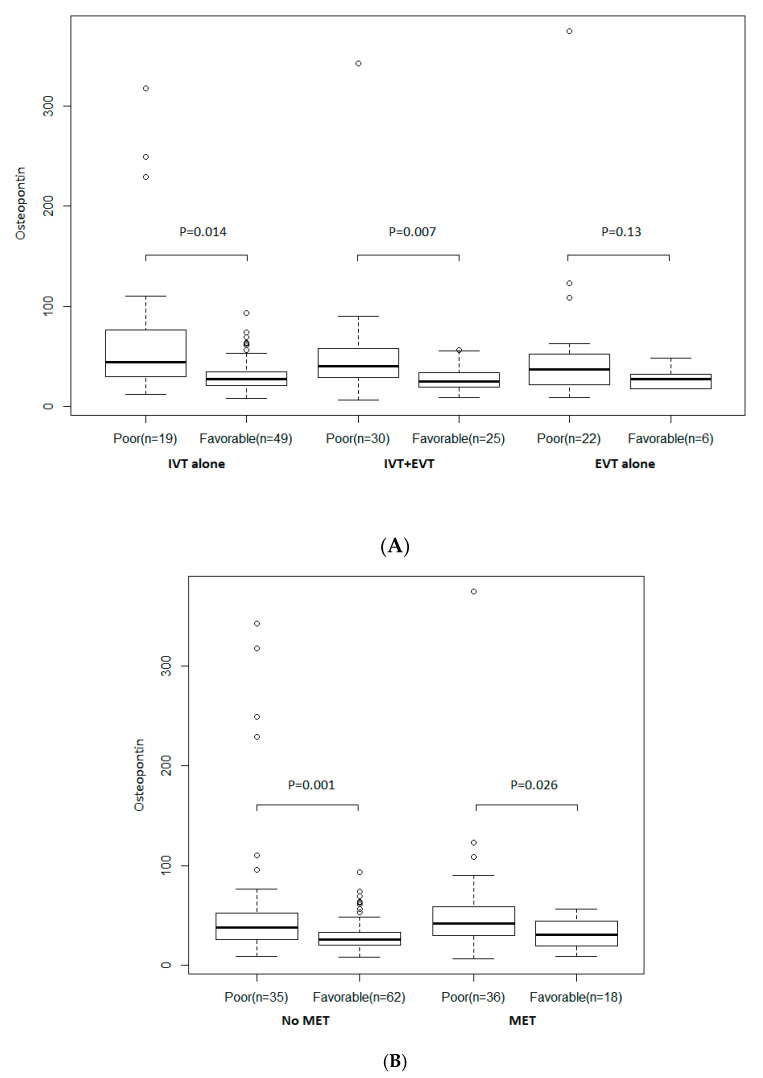
(**A**) According to treatment strategy. (**B**) According to mechanical endovascular treatment (MET). Distribution of osteopontin according to a favorable outcome and AIS treatment. Boxes show the 25th, 50th, and 75th percentiles, and unadjusted *p*-values (Student’s *t*-test on log-transformed values) are reported. Abbreviations: EVT = endovascular treatment; IV = intravenous thrombolysis; MET = mechanical endovascular treatment.

**Table 1 jcm-09-04028-t001:** Patients’ and treatment characteristics according to a favorable outcome.

	Favorable Outcome		
Characteristics	No (*n* = 71)	Yes (*n* = 80)	*p*-Value
Age, median (IQR)	78 (66 to 85)	64 (53 to 80)	<0.001
Men	35/71 (49.3)	43/80 (53.8)	0.58
Medical history			
Hypertension	46/71 (64.8)	45/78 (57.7)	0.37
Hypercholesterolemia	27/70 (38.6)	30/77 (39.0)	0.96
Diabetes	7/70 (10.0)	4/77 (5.2)	0.27
Current smoking	11/67 (16.4)	17/77 (22.1)	0.39
Antithrombotic medications	28/70 (40.0)	23/78 (29.5)	0.18
Antiplatelet	21/70 (30.0)	18/78 (23.1)	0.34
Anticoagulant	8/70 (11.4)	6/78 (7.7)	0.44
Pre-stroke mRs >1	34/71 (47.9)	12/78 (15.4)	<0.001
Current stroke event			
Admission Systolic BP, mmHg, mean (SD)	153 (22)	156 (18)	0.27
Admission Diastolic BP, mmHg, mean (SD)	80 (13)	83 (13)	0.17
Admission Glucose, mg/dl, mean (SD)	138 (40)	119 (34)	0.003
Admission Platelet counts, mean (SD)	220 (86)	231 (63)	0.40
Admission NIHSS, median (IQR)	18 (14 to 22)	8 (4 to 12)	<0.001
Documented arterial occlusion			
None	11/71 (15.5)	35/80 (44.9)	<0.001
ICA isolated or tandem with MCA	29/71 (40.8)	6/80 (7.7)	
MCA isolated	31/71 (43.7)	34/80 (43.6)	
Posterior circulation	0/71 (0.0)	3/80 (3.8)	
Cardioembolic etiology	39/71 (54.9)	33/78 (42.3)	0.12
Treatment characteristics			
AIS treatment			
IV thrombolysis	19/71 (26.8)	49/80 (61.3)	<0.001
Combined IV thrombolysis and Endovascular treatment (EVT)–	30/71 (42.3)	25/80 (31.3)	
EVT alone	22/71 (31.0)	6/80 (7.5)	
Mechanical endovascular treatment (MET)	36/71 (50.7)	18/80 (22.5)	<0.001
Onset to treatment time, min, median (IQR)			
IV alone	154 (120 to 227)	152 (120 to 187)	0.78
Combined IV–EVT	140 (115 to 160)	118 (90 to 145)	0.050
EVT alone	309 (230 to 350)	325 (250 to 345)	0.48

Values are number (%) unless otherwise as indicated. A favorable outcome is defined as 90-day mRS ≤ 2 or equal to pre-stroke mRs. Arterial occlusion documented before AIS treatment by transcranial Doppler, MRA or CT angiography. Abbreviations: AIS = acute ischemic stroke; BP = blood pressure; CT = computed tomography; EVT = endovascular treatment; IA = intra-arterial; ICA = internal carotid artery; IQR = interquartile range; IV = intravenous; MCA = middle cerebral artery; MET = mechanical endovascular treatment; MRA = magnetic resonance angiography; mRS = modified Rankin Scale; NIHSS = National Institutes of Health Stroke Scale; SD = standard deviation.

**Table 2 jcm-09-04028-t002:** Associations of biological markers with a favorable outcome.

	Favorable Outcome		Unadjusted	Adjusted ^2^	
Blood Biomakers, in ng/mL	No (*n* = 71)	Yes (*n* = 80)	*p*-Value ^1^	OR (95%CI)	*p*-Value ^1^
Hemostasis					
uPA/urokinase	1.18 (1.04 to 1.34)	1.11 (0.93 to 1.31)	0.58	0.63 (0.41 to 0.97)	0.61
Serpin E1/PAI-1	40 (20 to 67)	41 (21 to 66)	1.00	0.96 (0.60 to 1.55)	1.00
Serpin C1/antithrombin-III	4672 (2963 to 6149)	4444 (2175 to 5615)	1.00	0.78 (0.48 to 1.26)	1.00
Kallikrein 6/neurosin	4.7 (3.3 to 6.0)	4.8 (3.9 to 5.8)	1.00	1.00 (0.63 to 1.57)	1.00
Alpha2 macroglobulin	2541 (1030)	2414 (865)	1.00	0.97 (0.61 to 1.55)	1.00
Inflammation					
MPO	51 (37 to 74)	38 (28 to 50)	0.002	0.53 (0.31 to 0.89)	0.32
CCL2/MCP-1	0.26 (0.18 to 0.31)	0.24 (0.19 to 0.32)	1.00	0.96 (0.63 to 1.44)	1.00
Adiponectine	8689 (5118 to 14499)	7502 (5210 to 11821)	1.00	0.88 (0.54 to 1.41)	1.00
Resistin	9.6 (7.0 to 12.5)	8.3 (6.6 to 10.6)	1.00	0.85 (0.54 to 1.34)	1.00
cDNA	1.02 (0.93 to 1.13)	0.95 (0.86 to 1.04)	0.11	0.23 (0.08 to 0.66)	0.12
CD40 Ligand	0.99 (0.63 to 1.93)	0.88 (0.64 to 1.29)	1.00	0.80 (0.50 to 1.28)	1.00
Endothelial activation					
VCAM-1	1302 (834 to 1724)	838 (600 to 1215)	<0.001	0.53 (0.31 to 0.90)	0.32
ICAM-1	301 (225 to 511)	258 (192 to 389)	1.00	0.67 (0.40 to 1.10)	1.00
CD31/PECAM	9.7 (8.2 to 12.0)	9.7 (8.2 to 11.6)	1.00	1.01 (0.63 to 1.62)	1.00
Tissue remodeling					
Total cathepsin S	6.1 (4.3 to 8.4)	4.9 (3.6 to 6.6)	0.093	0.60 (0.37 to 0.98)	0.72
Osteopontin	40 (28 to 58)	26 (20 to 35)	0.001	0.34 (0.18 to 0.62)	0.008
Cystain C	825 (711 to 1024)	725 (619 to 885)	0.068	0.61 (0.36 to 1.05)	1.00
Neuropilin-1	229 (193 to 277)	196 (168 to 233)	0.011	0.64 (0.40 to 1.02)	0.98
MMP-2	253 (192 to 295)	238 (188 to 282)	1.00	1.34 (0.82 to 2.19)	1.00
MMP-3	8.6 (6.2 to 11.4)	9.1 (6.0 to 11.4)	1.00	0.95 (0.60 to 1.51)	1.00
MMP-9	153 (77 to 321)	112 (61 to 201)	0.24	0.67 (0.42 to 1.05)	1.00
MMP-13	0.33 (0.29 to 0.41)	0.37 (0.32 to 0.44)	1.00	0.86 (0.54 to 1.37)	1.00

Values are median (IQR) except for Alpha2 macroglobulin, where mean (SD) is reported. For all biological markers (except Alpha2 macroglobulin), *p*-values and ORs were calculated on log-transformed values. ORs were calculated per one SD increase in biological markers. A favorable outcome is defined as 90-day mRS ≤ 2 or equal to pre-stroke mRS. ^1^ corrected for multiple comparisons using Holm Bonferroni method. ^2^ adjusted for age, admission NIHSS, documented occlusion (none vs. ICA occlusion vs. others arterial occlusion), and AIS treatments (including IV+-EVT approach). Abbreviations: AIS = acute ischemic stroke; CI = confidence interval; IV Intra-venous, EVT Endovascular treatment; ICA = internal carotid artery; IQR = interquartile range; IV = intravenous; mRS = modified Rankin Scale; OR = odds ratio; SD = standard deviation.

**Table 3 jcm-09-04028-t003:** Associations of biological markers with a excellent outcome.

	Excellent Outcome		Unadjusted	Adjusted ^2^	
Blood Biomarkers, in ng/mL	No (*n* = 87)	Yes (*n* = 64)	*p*-Value ^1^	OR (95%CI)	*p*-Value ^1^
Hemostasis					
uPA/urokinase	1.18 (1.02 to 1.34)	1.08 (0.90 to 1.29)	0.27	0.61 (0.38 to 0.98)	0.67
Serpin E1/PAI-1	39 (19 to 65)	42 (22 to 67)	1.00	1.14 (0.74 to 1.75)	1.00
Serpin C1/antithrombin-III	4481 (2653 to 5581)	4567 (2951 to 5870)	1.00	1.12 (0.74 to 1.71)	1.00
Kallikrein 6/neurosin	4.6 (3.5 to 5.7)	4.9 (4.1 to 5.9)	1.00	1.19 (0.77 to 1.81)	1.00
Alpha2 macroglobulin	2459 (986)	2494 (805)	1.00	1.23 (0.81 to 1.86)	1.00
Inflammation					
MPO	49 (35 to 71)	37 (27 to 50)	0.009	0.60 (0.37 to 0.97)	0.62
CCL2/MCP-1	0.25 (0.18 to 0.32)	0.23 (0.19 to 0.30)	1.00	0.92 (0.62 to 1.37)	1.00
Adiponectine	8100 (4226 to 14499)	7619 (5363 to 11781)	1.00	1.05 (0.68 to 1.61)	1.00
Resistin	9.1 (6.7 to 11.8)	8.3 (6.6 to 10.7)	1.00	1.08 (0.72 to 1.61)	1.00
cDNA	1.02 (0.93 to 1.12)	0.95 (0.85 to 1.03)	1.00	0.76 (0.49 to 1.17)	1.00
CD40 Ligand	0.91 (0.63 to 1.93)	0.89 (0.64 to 1.24)	0.37	0.65 (0.41 to 1.02)	0.91
Endothelial activation					
VCAM-1	1237 (826 to 1682)	821 (592 to 1040)	<0.001	0.54 (0.33 to 0.89)	0.29
ICAM-1	309 (228 to 530)	228 (177 to 337)	0.27	0.59 (0.37 to 0.92)	0.37
CD31/PECAM	9.9 (8.3 to 12.1)	9.5 (8.0 to 11.1)	1.00	0.74 (0.47 to 1.15)	1.00
Tissue remodeling					
Total cathepsin S	5.9 (4.3 to 8.2)	4.5 (3.5 to 6.5)	0.048	0.58 (0.37 to 0.91)	0.36
Osteopontin	38 (24 to 55)	26 (20 to 33)	0.010	0.49 (0.29 to 0.81)	0.12
Cystain C	812 (694 to 1014)	753 (604 to 938)	0.68	0.89 (0.55 to 1.44)	1.00
Neuropilin-1	226 (188 to 274)	191 (167 to 233)	0.048	0.71 (0.45 to 1.09)	1.00
MMP-2	252 (193 to 295)	235 (181 to 279)	1.00	1.19 (0.74 to 1.89)	1.00
MMP-3	8.6 (6.1 to 11.4)	9.1 (6.5 to 11.4)	1.00	1.03 (0.67 to 1.56)	1.00
MMP-9	138 (67 to 264)	120 (63 to 209)	1.00	0.90 (0.60 to 1.34)	1.00
MMP-13	0.34 (0.29 to 0.42)	0.37 (0.29 to 0.42)	1.00	0.83 (0.52 to 1.32)	1.00

Values are median (IQR) except for Alpha2 macroglobulin, where mean (SD) is reported. For all biological markers (except Alpha2 macroglobulin), *p*-values and ORs were calculated on log-transformed values. ORs were calculated per one SD increase in biological markers. An excellent outcome is defined as 90-day mRS ≤ 1 or equal to pre-stroke mRS. ^1^ corrected for multiple comparisons using Holm Bonferroni method. ^2^ adjusted for age, admission NIHSS, documented occlusion (none vs. ICA occlusion vs. others arterial occlusion), and AIS treatments (including IV +-EVT). Abbreviations: AIS = acute ischemic stroke; CI = confidence interval; IV Intra-venous, EVT Endovascular treatment; ICA = internal carotid artery; IQR = interquartile range; IV = intravenous; MET = mechanical endovascular treatment; mRS = modified Rankin Scale; OR = odds ratio; SD = standard deviation.

**Table 4 jcm-09-04028-t004:** Associations of biological markers with early neurological improvement.

	Early Neurological Improvement		Unadjusted	Adjusted ^2^	
Blood Biomarkers, in ng/mL	No (*n* = 79)	Yes (*n* = 72)	*p*-Value ^1^	OR (95%CI)	*p*-Value ^1^
Hemostasis					
uPA/urokinase	1.15 (1.01 to 1.36)	1.15 (0.95 to 1.29)	1.00	0.98 (0.67 to 1.41)	1.00
Serpin E1/PAI-1	40 (23 to 67)	41 (20 to 66)	1.00	0.84 (0.57 to 1.22)	1.00
Serpin C1/antithrombin-III	4693 (2930 to 6055)	4314 (2881 to 5639)	1.00	0.83 (0.56 to 1.23)	1.00
Kallikrein 6/neurosin	4.8 (3.6 to 5.8)	4.8 (3.8 to 6.0)	1.00	0.87 (0.58 to 1.29)	1.00
Alpha2 macroglobulin	2571 (1036)	2367 (829)	1.00	0.80 (0.54 to 1.17)	1.00
Inflammation					
MPO	49 (37 to 73)	37 (27 to 52)	0.005	0.51 (0.32 to 0.80)	0.064
CCL2/MCP-1	0.22 (0.16 to 0.31)	0.25 (0.20 to 0.31)	1.00	1.11 (0.77 to 1.59)	1.00
Adiponectine	8373 (5273 to 14587)	7454 (5051 to 11788)	1.00	0.86 (0.58 to 1.25)	1.00
Resistin	9.2 (7.0 to 12.5)	7.7 (6.4 to 10.6)	0.83	0.70 (0.47 to 1.05)	1.00
cDNA	0.98 (0.92 to 1.10)	0.96 (0.88 to 1.08)	1.00	1.23 (0.83 to 1.81)	1.00
CD40 Ligand	0.98 (0.65 to 1.65)	0.86 (0.60 to 1.38)	1.00	0.82 (0.56 to 1.20)	1.00
Endothelial activation					
VCAM-1	1034 (730 to 1641)	908 (663 to 1405)	1.00	0.98 (0.66 to 1.46)	1.00
ICAM-1	309 (225 to 571)	255 (192 to 337)	1.00	0.76 (0.52 to 1.09)	1.00
CD31/PECAM	9.7 (8.3 to 11.9)	9.8 (8.0 to 11.6)	1.00	0.87 (0.60 to 1.27)	1.00
Tissue remodeling					
Total cathepsin S	6.0 (4.4 to 8.4)	4.4 (3.6 to 6.5)	0.007	0.54 (0.35 to 0.81)	0.064
Osteopontin	32 (23 to 52)	30 (20 to 45)	1.00	0.80 (0.55 to 1.17)	1.00
Cystain C	812 (666 to 958)	758 (651 to 975)	1.00	0.89 (0.57 to 1.36)	1.00
Neuropilin-1	227 (188 to 268)	203 (170 to 236)	0.55	0.76 (0.52 to 1.11)	1.00
MMP-2	248 (192 to 282)	242 (192 to 292)	1.00	1.30 (0.86 to 1.97)	1.00
MMP-3	8.1 (5.8 to 11.3)	9.3 (7.1 to 12.4)	0.43	1.60 (1.09 to 2.34)	0.29
MMP-9	159 (85 to 290)	97 (57 to 201)	0.059	0.62 (0.42 to 0.90)	0.23
MMP-13	0.34 (0.29 to 0.41)	0.38 (0.31 to 0.44)	1.00	1.03 (0.69 to 1.53)	1.00

Values are median (IQR) except for Alpha2 macroglobulin, where mean (SD) is reported. For all biological markers (except Alpha2 macroglobulin), *p*-values and ORs were calculated on log-transformed values. ORs were calculated per one SD increase in biological markers. An early neurological outcome is defined as 90-day mRS ≤ 1 or equal to pre-stroke mRS. ^1^ corrected for multiple comparisons using Holm Bonferroni method. ^2^ adjusted for age, admission NIHSS, documented occlusion (none vs. ICA occlusion vs. others arterial occlusion), and AIS treatments (including IV+-EVT). Abbreviations: AIS = acute ischemic stroke; CI = confidence interval; IV Intra-venous, EVT Endovascular thrombectomy; ICA = internal carotid artery; IQR = interquartile range; IV = intravenous; MET = mechanical endovascular treatment; mRS = modified Rankin Scale; OR = odds ratio; SD = standard deviation.

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
