# Peer review of "Osteopontin Predicts Three-Month Outcome in Stroke Patients Treated by Reperfusion Therapies"

_jcm, 2020, doi:10.3390/jcm9124028_

Round 1
Reviewer 1 Report
In this paper the authors evaluated biomarkers associated with favorable outcome after stroke treated by reperfusion therapy.
The authors raised a very siginificant issue- to find a biomarker on admission enable to predict the clinical outcome.
The authors found that osteopontin , related to tissue remodeling, could be such a factor.
The manuscript is well-organized, with scientific soundness.
No significant drawbacks or flaws have been found.
Some issues should be discussed:
it is mandatory to provide a number of Bioethics Committee approval,
please explain why only stroke subjects who underwent reperfusion therapy were included? it could be more interesting to compare the role of biomarkers in stroke subjects undergoing reperfusion therapy and standard antiplatelet treatment without specific treatment.It could underline the role of specific treatment for this association.
the authors analyzed general prediction in all subjects, both after thrombolysis and after thrombolysis with thrombectomy. it could be very interesting if ostepontin is a predictor in subjects underwent only thrombolysis without thrombectomy? Is it possible to show the results in both group of selected subjects? this could improve the quality of the paper.
Author Response
Dear colleagues,
Thanks for the remarks that would improve the quality of the article. Here are our comments:
- The Bioethics Committee approval is the Ambroise Pare Hospital, on September 23rd 2008, code ID RCB: 2008-A00966-49 and January 05th 2010, code ID RCB: 2009-A01165-52, Paris, France. It has been added in the manuscript (SECTION METHODS, line 85-87).
- Reperfusion therapies have a clear impact in stroke prognosis. This is a first study and we wanted to focus in a homogenous group of patients to generate hypothesis and select those biomarkers that could be relevant. A new project including patients under antiplatelet treatment and reperfusion therapies is going nowadays. This limitation has been added in the last paragraph of the discussion, lines 340-341.
- Figure 2 A showed Osteopontin levels according to the treated received. In patients under exclusively intravenous thrombolysis and those treated by intravenous thrombolysis + intra-arterial thrombolysis or thrombectomy, Osteopontin was related to poor outcome, p=0.014 and p=0.007 respectively. In those under only intra-arterial thrombolysis or thrombectomy there was no difference, but only 6 patients with favorable outcome were included in the analyses. We will add a commentary in the paper according to the treatment received. A paragraph explaining it has been added at the end of the results, lines 274-279 ; instead of 224-226
Reviewer 2 Report
The article is focused on a very hot-topic issue and it is well designed and performed. I have some points to imporve the article:
- Please include a table summarizing the ELISA kit used, reporting the lower range limit, and product reference.
- In previous studies, CCL2 was reported to reach very small concentrations, around ten-fold less than reported in the study. Can the authors check and discuss these results.
- Which was the timing of blood sampling? Thrombolysis might impact in a time depedent manner mainly on MMPs concentrations.
- Did authors used serum or plasma? The levels of some markers can be differently impacted on the different human samples.
Author Response
Dear colleagues,
Thanks for the remarks that would improve the quality and the precision of the article.
Here are our answers:
- Four differents human Magnetic Luminex assay kits were used (reference LXSAHM, R&D system Bio- Techne as specified in materiel and method section line 127-128) due to the fact that different dilutions were required and because of biological incompatibilities between analytes according to the manufacturer’s instruction.
The lower range limit varies for the analytes as follow (in ng/ml):
u-Plasminogen Activator (uPA)/Urokinase: 0,12
Serpin E1/PAI-1: 0,02
Serpin C1/Antithrombin-III: 2,84
Cathepsin S: 0,09
CCL2/JE/MCP-1 : 0,03
CD31/PECAM-1: 0,68
CD40 Ligand/TNFSF: 0,21
ICAM-1/CD54: 6,89
Kallikrein 6/Neurosin: 0,28
MMP-3: 0,08
MMP-13: 0,12
Osteopontin/OPN: 1,8
Resistin: 0,04
VCAM-1/CD106 : 7,80
alpha 2-Macroglobulin: 1,66
Adiponectin: 0,95
Cystatin C: 0,66
MMP-9: 0,13
Neuropilin-1: 0,08
MMP-2: 0,32
MPO: 0,13
That information was added in blood markers section line 120 until 125.
- As showed by Scholman et al. (Cytokine, 2018), in a study where they compared 162 circulating proteins either from serum, plasma heparin or plasma EDTA from healthy donors, it is true that for a same analyte variations in the dosage can be found. It could be influenced by different methodological factors, i.e. the type of anticoagulant used, the kit used to carry out the dosage, conservation... Nonetheless, this study conducted on 43 healthy donors (11 males and 32 females) show similar amount of CCL2 as in our hands and as other studies (Lubowicka et al., 2018, BioMed Research International; Westin et al., 2012, Plos One; Gu et al., 2010, Human Immunology).
- Blood was sampled within 6 hours after stroke onset and before any reperfusion treatment was given. We added this information in the methods, part Blood Biomarkers, line 102-103
4 Levels of biomarkers were measured in plasma. This information is found in the methods section (lines 112,127) and now has been added in the first paragraph of the discussion, lines 289, 291.
As correctly pointed by the reviewer, the type of blood sampling can modify levels of measured analytes. We have now indicated that our blood was collected on EDTA tubes (added in text lines 112, 127, 129, 289 and 291).
Round 2
Reviewer 2 Report
No further comments.